# A Synthetic Formula Amino Acid Diet Leads to Microbiome Dysbiosis, Reduced Colon Length, Inflammation, and Altered Locomotor Activity in C57BL/6J Mice

**DOI:** 10.3390/microorganisms11112694

**Published:** 2023-11-03

**Authors:** Viviana J. Mancilla, Paige N. Braden-Kuhle, Kelly N. Brice, Allison E. Mann, Megan T. Williams, Yan Zhang, Michael J. Chumley, Robert C. Barber, Sabrina N. White, Gary W. Boehm, Michael S. Allen

**Affiliations:** 1Department of Microbiology, Immunology, and Genetics, School of Biomedical Sciences, University of North Texas Health Science Center, Fort Worth, TX 76107, USA; 2Department of Psychology, College of Science and Engineering, Texas Christian University, Fort Worth, TX 76109, USA; 3Department of Biological Sciences, College of Science, Clemson University, Clemson, SC 29634, USA; 4Department of Biology, College of Science and Engineering, Texas Christian University, Fort Worth, TX 76109, USA; m.chumley@tcu.edu; 5Department of Pharmacology and Neuroscience, School of Biomedical Sciences, University of North Texas Health Science Center, Fort Worth, TX 76107, USA

**Keywords:** SCFA, inflammation, mouse model, fiber, synthetic diet, gut-brain axis, butyrate, prebiotics

## Abstract

The effects of synthetic, free-amino acid diets, similar to those prescribed as supplements for (phenylketonuria) PKU patients, on gut microbiota and overall health are not well understood. In the current, multidisciplinary study, we examined the effects of a synthetically-derived, low-fiber, amino acid diet on behavior, cognition, gut microbiome composition, and inflammatory markers. A cohort of 20 male C57BL/6J mice were randomly assigned to either a standard or synthetic diet (*n* = 10) at post-natal day 21 and maintained for 13 weeks. Sequencing of the 16S rRNA gene from fecal samples revealed decreased bacterial diversity, increased abundance of bacteria associated with disease, such as *Prevotella*, and a downward shift in gut microbiota associated with fermentation pathways in the synthetic diet group. Furthermore, there were decreased levels of short chain fatty acids and shortening of the colon in mice consuming the synthetic diet. Finally, we measured TNF-α, IL-6, and IL-10 in serum, the hippocampus, and colon, and found that the synthetic diet significantly increased IL-6 production in the hippocampus. These results demonstrate the importance of a multidisciplinary approach to future diet and microbiome studies, as diet not only impacts the gut microbiome composition but potentially systemic health as well.

## 1. Introduction

The gut microbiome plays a variety of fundamental roles in health and disease, including protection against opportunistic pathogens, extracting nutrients and energy from the diet, and helping maintain normal immune function [1,2,3,4,5]. The gut microbiome also impacts distal organs and systems, such as the nervous and immune systems, via microbial signaling and metabolites, including short chain fatty acids (SCFAs) [6,7,8,9,10,11,12,13,14]. Furthermore, gut microbes can affect neurochemistry via stimulation of vagal afferents and/or the production of SCFAs and choline, which are both important for brain health [15]. Previous research has demonstrated that a maternal diet high in fiber protects offspring from maternal-obesity-induced cognitive dysfunction [16].

Factors including age, genetics, pharmaceutical usage (e.g., antibiotics), and diet can significantly alter gut microbiome composition [17]. Mammalian gut microbiomes share a large core repertoire of functions, such as amino acid metabolism and biosynthetic pathways. However, environmental stimuli, particularly diet, can result in microbiomes that are specialized to degrade and metabolize compounds commonly found in that diet [18]. Sudden changes in diet have been demonstrated to rapidly shift the community structure of the microbiota, change the representation of metabolic pathways in the microbiome, and alter microbiome gene expression [19]. Unraveling the complex interrelationships between diet, health, and the gut microbiota undoubtedly holds major implications for understanding and treating disease, and numerous questions remain unanswered.

Dietary fiber has been shown to be a primary driver of gut microbiome composition [19,20,21]. Nutritionists recommend that adults consume about 25–35 g of fiber per day, though the average American only consumes about 15 g [22]. The typical American diet, often designated as the Western diet, is a fiber-deficient diet mostly comprised of simple carbohydrates and animal-based sources of fat and protein. The Western diet is rich in saturated fatty acids, refined sugars, and carbohydrates, yet simultaneously deficient in plant-based fats and proteins. The Western diet provides about 50% kcal from carbohydrates, such as refined carbohydrates and grains, per day [23]. The fiber-deficient Western diet is associated with several chronic diseases, including obesity, diabetes mellitus type II, dementia, inflammatory bowel disease, and ulcerative colitis, all of which entail significant chronic inflammation. Prior research demonstrates that consumption of a Western-style, low-fiber diet is associated with increased inflammation [24,25]. Temba et al. (2021) studied the effects of a plant-based, high-fiber diet versus a Western-style, low-fiber diet on peripheral inflammation in adults, and found that the Western-style diet exacerbated the production of pro-inflammatory mediators, such as TNF-α and IL-6, upon immune activation [26]. Conversely, plant-based diets, such as the Mediterranean diet, are rich in fiber and provide gut microbiota with important nutrients which support the production of anti-inflammatory compounds [27]. 

Although generally indigestible to humans, dietary fiber is a source of microbiota-accessible carbohydrates (MACs). These MACs transit the small intestine to the colon where they serve as substrates for strictly anaerobic fermentative bacteria that subsequently produce short-chain fatty acids (SCFAs) such as butyrate, propionate, and acetate [28]. These SCFAs are then metabolized by colonocytes in a way that results in high oxygen consumption, thus maintaining epithelial hypoxia and favoring the growth of similar commensal, obligately anaerobic SCFA producers [29]. Low fiber consumption starves colonic bacteria of complex carbohydrates, which in turn decreases microbiome diversity and increases the risk of developing dysbiotic host-microbiome interactions [30] This shift adversely impacts the intestinal mucosal barrier. For instance, findings demonstrate that a low-fiber diet promotes the expansion and activity of a subset of colonic bacteria that act to erode the colonic mucosal barrier, increasing the risk of opportunistic infections [31]. Furthermore, low-fiber diets fail to support SCFA producers in the gut, thus starving gut epithelial cells of these energy sources and resulting in higher levels of anaerobic glycolysis, a process characterized by high lactate release and low oxygen consumption. Under these conditions, epithelial hypoxia cannot be maintained, and facultative anaerobes begin to flourish in the gut, resulting in dysbiotic interactions and inflammation [29]. Overall, diet not only influences the composition of the gut microbiome but also affects microbial metabolism, which in turn profoundly influences host health [29].

Several groups have linked microbiota-generated SCFAs to anti-inflammatory effects. Indeed, SCFAs increase the pool of regulatory T-cells in the gut and are protective against allergic airway inflammation in a mouse model [6,32]. In a study examining ulcerative colitis patients, mucosal levels of pro-inflammatory cytokines were reduced by administration of butyrate, which correlated with a decrease in disease pathology [33]. Additionally, and particularly relevant to the current study, children diagnosed with phenylketonuria (PKU) were found to have decreased levels of fecal butyrate associated with a restrictive, low-phenylalanine diet, along with phenylalanine-free amino acid formula supplementation, despite consuming soluble fiber [34]. Children suffering from PKU were compared to children diagnosed with mild hyperphenylalaninemia (MHP), and while MHP children are affected by PKU, they are not on a phenylalanine-restrictive diet. The authors also describe the childrens’ gut microbiota as deficient in *Faecalibacterium prausnitzii* and *Roseburia* species. *F. prausnitzii* abundance has previously been shown to be positively associated with fecal butyrate content [35]. Similarly, a previous study also reported that adults diagnosed with PKU have significantly decreased abundance of butyrogenic bateria, specifically *Faecalibacterium*, compared to healthy control individuals [36].

The effects of heavily synthetic, phenylalanine-restrictive diets on patients, their gut microbiomes, and overall health are not well understood. In the current study, we aimed to elucidate the impact of a synthetic, free-amino acid diet analogous to that prescribed as supplementation for PKU patients (but without specific phenylalanine restriction) on the gut microbiome, inflammation, and behavior in wild-type mice. Specifically, two groups of mice were fed either a diet of standard rodent chow or a synthetic diet version, in which all complex proteins were replaced with equivalent levels of free amino acids for a period of 13 weeks. The diets were matched as closely as possible for macronutrient content with differences in the nutritional sources, leading to decreased insoluble fiber and lack of soluble fiber content (i.e., MACs) in the synthetic diet (additional nutritional information can be found in Appendix A). 

Over the course of the study, the bacterial composition of the gut microbiome was assessed, along with potential alterations in behavior. Finally, multiple tissues were assayed to assay for changes in markers of inflammation and SCFAs. We found that mice consuming the synthetic diet showed significantly altered gut bacterial microbiome profiles. Furthermore, these changes suggest systemic inflammation, as evidenced by alterations in cytokine profiles in the brain and dramatic differences in colon length. These results are similar to those frequently reported from studies on the effects of a Western diet, although the synthetic diet used in the current study did not include animal protein or increased fat [37]. Taken together, the results here suggest that dietary fiber, specifically soluble and insoluble fiber, may play a larger role in health than is currently understood.

## 2. Methods and Materials

### 2.1. Experimental Design

Male C57BL/6J mice bred in the Texas Christian University vivarium from a breeding stock obtained from The Jackson Laboratory (Bar Harbor, ME, USA) were utilized for this experiment. All mice received care according to the Guide for the Care and Use of Laboratory Animals (National Research Council, 1996), and all study protocols were approved by the Texas Christian University Institutional Animal Care and Use Committee (TCU IACUC #19/002). Mice were housed under a 12-h light/dark cycle (lights off period beginning at 1900 h), with 3–4 mice per standard polycarbonate cage in each group (30 × 20 × 16 cm). Animals were housed in cages with pelleted, paper-chip bedding and compressed cotton square nestlets (Lab Supply, Fort Worth, TX, USA). The vivarium housing rooms were kept at a temperature between 19 °C and 23 °C and humidity between 40 and 60%. Food and water were available ad libitum. All mice were weaned at postnatal day 21. Following weaning, animals were randomly assigned to one of two diet conditions, including a standard rodent diet (*n* = 10), Prolab RMH 1800 (LabDiet, St. Louis, MO, USA), or the synthetic formula diet (*n* = 10), TD 190275 Amino Acid Diet (5LL2) (Envigo Teklad Diets, Madison, WI, USA) for 13 weeks (Figure 1). We matched the macronutrient profile of the synthetic diet as closely as possible to that of standard rodent diet (Appendix A). However, while the standard mouse diet derives its nutrients from whole food ingredients (e.g., ground corn, wheat middlings, dehulled soybean meal, and ground wheat), the synthetic diet is primarily comprises single amino acids, synthetically derived vitamins, and simple fat and carbohydrate sources (Appendix A). Crude fiber percentage was also matched between the standard and synthetic diets despite the difference in nutritional sources (Appendix A). However, the soluble and insoluble fiber content is decreased and depleted, respectively, in the synthetic diet (Appendix A). While fiber sources for the standard diet comprised whole food ingredients, fiber sources for the synthetic diet were primarily cellulose and corn starch, the former being largely unavailable to gut microbes. 

Mouse fecal samples were collected weekly for gut microbiome analysis to monitor changes in bacterial diversity over the course of the feeding period, as well as differential taxa abundance and hypothesized metabolic pathway alterations at the conclusion of the feeding period. Fecal samples were collected weekly to better understand how soon after starting one of two diet conditions changes begin to be observed and how these trends change longitudinally. Following 13 weeks of diet consumption, behavioral tests were then conducted, including an open-field test, elevated zero maze, and contextual fear conditioning. Mouse colon, blood, and hippocampal tissue were then harvested, snap-frozen in dry ice, and stored at −80 °C until further processing. Subsequent analyses of tissues examined the effects of consuming a synthetic diet on colon length, cytokine levels, and SCFA levels. Details of experimental protocols are provided in Figure 1.

### 2.2. Microbiome Sample Collection

Mice were briefly separated on a weekly basis into individual clean cages, without bedding, to easily collect fecal pellets. Samples were collected in individual sterile 1.5 mL microcentrifuge tubes, and immediately placed on dry ice. After collection, fecal pellets were transferred and stored at −80 °C until further processing. 

### 2.3. DNA Extraction and 16S rRNA Gene Amplification

DNA was extracted from mouse fecal samples by following the manufacturer’s protocol for the Kingfisher MagMAX Microbiome Ultra Nucleic Acid Isolation Kit (ThermoFisher Scientific, Waltham, MA, USA). Extracted DNA was eluted to an adjusted final volume of 50 μL, quantified using a NanoDrop 1000 spectrophotometer (ThermoFisher Scientific Inc., Waltham, MA, USA), and stored at −20 °C.

Barcoding polymerase chain reaction (PCR) was performed, followed by indexed primers targeting the V4 hypervariable region of the 16S rRNA gene, utilizing illumv4_515F (5′-GTGCCAGCMGCCGCGGTAA-3′) and illumv4_806R (5′-GGACTACHVGGGTWTCTAAT-3′) primers with Illumina sequencing adaptors [38]. PCR reactions were prepared in duplicate at 25 μL volumes consisting of 2.5 μL 10× Accuprime™ PCR Buffer II (Invitrogen, Carlsbad, CA, USA), 0.5 μL forward and reverse primer (10 μM), 0.1 μL of Accuprime™ Taq DNA Polymerase High Fidelity (5 U/μL), 1 μL of template DNA (10–100 ng), and 16.9 μL molecular grade water. Negative and positive controls were produced along with every master mix, containing either 1-μL *Escherichia coli* genomic DNA or 1 μL of molecular grade water, respectively. PCR was carried out as follows: 94 °C for 2 min, 30 cycles of 94 °C for 30 s, 55 °C for 40 s, 68 °C for 40 s, and a final extension at 68 °C for 5 min. Successful PCR was confirmed by electrophoresis in 1% agarose gels. Bands were observed in all positive PCR controls and absent for all negative controls. 

### 2.4. Library Preparation and Sequencing

Duplicate PCR products were pooled and purified using Agencourt^®^ AMPure^®^ XP magnetic beads (Beckman Coulter, Brea, CA, USA), following the manufacturer’s instruction. Multiplexing was possible through dual indexing, using the Nextera XT assay on purified amplicons. Briefly, 50 μL PCR volumes were prepared for each sample with the following recipe and thermocycler protocol: 5 μL 10× Accuprime™ PCR Buffer II, 5 μL Nextera XT Index Primer 1, 5 μL Nextera XT Index Primer 2, 0.2 μL, Accuprime™ Taq DNA Polymerase High Fidelity, 5 μL DNA (~10 pm), and 29.8 μL of molecular grade water; 94 °C for 3 min, 8 cycles of 94 °C for 30 s, 55 °C for 30 s, 68 °C for 30 s, and a final extension at 68 °C for 5 min.

Once indexed, PCR products were purified as previously described, using AMPure^®^ XP magnetic beads, and quantified using the Qubit^®^ 2.0 fluorometer (Invitrogen, Carlsbad, CA, USA). Finally, libraries were pooled at equimolar concentrations, then diluted and denatured for a final concentration of 10 pM. Pooled samples were then loaded into a MiSeq Reagent Kit v2 cartridge (Illumina Inc, San Diego, CA, USA), and sequenced on an Illumina MiSeq^®^ instrument, along with an internal control sample consisting of 5% PhiX DNA, for paired-end high-throughput sequencing at 500 cycles.

### 2.5. Controlling for Contamination

In an effort to control for environmental contamination, blanks and negative controls were run alongside samples through the entire process, from extraction through sequencing. Blank extractions read below the quantifications limit. Negative controls and blanks amplified in the same 96-well plates, as experimental samples did not yield a band in gel electrophoresis. Blank libraries were constructed and read below quantification limits.

### 2.6. Data Processing

FASTQ files generated by the Illumina MiSeq^®^ instrument were utilized for bioinformatic processing and microbiome analysis. Sequencing primer reads were trimmed from raw reads, using Cutadapt v. 1.16 [39]. Forward and reverse reads were trimmed 50 bp from the 3′ end. Trimmed forward reads shorter than 150 bp and reverse reads shorter than 120 bp were discarded, and remaining reads were quality filtered, merged, checked for chimeras, and clustered into Amplicon Sequencing Variants (ASVs) using the DADA2 pipeline [40]. Processed sequences were then assigned taxonomy, using VSEARCH v. 2.8.1 and the EzBiocloud database as a reference [41,42]. Data analysis was performed as previously described by Mann et al. in 2020, primarily within the R version 3.5.0 environment [41,43,44]. Sequence data were deposited to the NCBI BioProject database under BioProject ID PRJNA908095. 

### 2.7. Microbiome Analysis

Alpha diversity (Shannon diversity index and observed ASV counts) was calculated using phyloseq v. 1.30.0 [45]. Beta diversity was determined through Principal Component Analysis (PCA) using Phylogenetic isometric log-ratio (PhILR) transformed data and the Vegan v. 2.5–5 library [46,47]. Significance of variance found through PCA was determined via permutational multivariate analysis of variance (PERMANOVA) [48]. 

Differential abundance of ASVs and phylogenetic clades between dietary groups was calculated with DESeq2 and Phylofactor [49,50]. Additionally, the effect of synthetic diet consumption on the metabolic potential of gut bacteria was predicted through the PICRUSt2 (Phylogenetic Investigation of Communities by Reconstruction of Unobserved States) pipeline [51]. PICRUSt2 results were visualized with STAMP (STatistical Analysis of Metagenomic Profiles) [52]. Scripts for all read processing, analyses, and visualization can be accessed at https://github.com/vivmancilla/SyntheticDiet.

### 2.8. Tissue Collection

Following the behavioral testing described below, all mice were humanely euthanized via rapid decapitation. Trunk blood was collected and placed on wet ice for 15 min, followed by incubation at room temperature for 30 min, then centrifugation (2000× *g*) at 4 °C for 10 min. Serum was isolated and immediately stored at −80 °C until cytokine analysis. Colon tissue, including the colon contents, and the hippocampus from both hemispheres of the brain were carefully dissected from the mice. Colons were measured from the intersection between the cecum and the colon to the anus. Tissues were lysed, preserved in a protein extraction solution containing protease inhibitors (PRO-PREP, Bulldog Bio, Portsmouth, NH, USA), snap-frozen on dry ice, and stored at −80 °C until processing. Tissue samples were centrifuged (16,820× *g*) at 4 °C for 20 min and, finally, clear lysate was extracted and stored at −20 °C until cytokine analysis. 

### 2.9. Short Chain Fatty Acid Analysis

Short chain fatty acids (SCFAs) in the colon were quantified by Microbiome Insights Inc. (Vancouver, BC, Canada) through gas chromatography (GC), following a similar protocol as in Zhao et al. (2006). Colon tissue was flash frozen and shipped on dry ice to Microbiome Insights Inc. (Vancouver, BC, Canada). The material was resuspended in MilliQ-grade H_2_O, and homogenized using MP Bio FastPrep, for 1 min at 4.0 m/s. Five M HCl was added to acidify fecal suspensions to a final pH of 2.0. Acidified fecal suspensions were incubated and centrifuged at 10,000 RPM to separate the supernatant. Fecal supernatants were spiked with 2-Ethylbutyric acid for a final concentration of 1 mM.

Extracted SCFA supernatants were stored in 2-mL GC vials with glass inserts. SCFA were detected using gas chromatography (Thermo Trace 1310), coupled to a flame ionization detector (ThermoFisher Scientific, Waltham, MA, USA). SCFAs were then detected via direct injection of supernatant into a ‘Thermo TG-WAXMS A GC Column (30 m, 0.32 mm, 0.25 µm). Standard solutions of individual SCFAs were used for calibration.

### 2.10. Cytokine Analysis

Concentrations of proinflammatory cytokines TNF-α and IL-6 and the anti-inflammatory cytokine IL-10 were quantified in the serum, hippocampal tissue lysates, and colon tissue lysates to assess inflammation, using VPLEX Custom Mouse Cytokine Proinflammatory Panel 1 (mouse) multiplexing kits (Meso Scale Diagnostics, Rockville, MD, USA). All wells in the provided plate were washed three times with a wash buffer before samples were plated. Serum samples were diluted 1:1 in the plates with a proprietary diluent, while hippocampal and colon samples were plated neat. Samples were allowed to incubate at room temperature with horizontal shaking at 750 rpm for two hours. All wells were then washed three times with wash buffer, and a detection antibody solution (for TNF-α, IL-6, and IL-10) was added to each well, followed by a two-hour incubation period at room temperature, with shaking. All wells were washed again three more times with wash buffer. Read buffer was then added to each well, and the electrochemiluminescent signal was read using a QuickPlex SQ 120 instrument (Meso Scale Diagnostics, Rockville, MD, USA). Samples were excluded from analysis if their values fell outside of the standard curve of the assay. 

### 2.11. Open Field

All behavioral testing occurred between 0800 h and 1200 h local time. The open field test (reviewed in [53]) was utilized to measure anxiety-like behavior, exploratory behavior, and locomotor activity in male C57BL/6J mice, following 13 weeks of diet consumption. During testing, mice were placed into individual, sound-attenuating open field maze chambers (27 × 27 cm) that were equipped with video tracking software (Med Associates Incorporated, St. Albans, VT, USA) to record movement. The software divides the open field chamber into two separate zones, including the center zone (the center of the chamber) and the outer zone (the remaining space around the center zone). Animals were individually removed from their home cages, gently placed in the center zone of the open field chamber, and allowed to freely explore for 10 min. Generally, the center zone is more aversive to mice, so more time spent in the center zone suggests reduced anxiety-like behavior compared to spending more time in the outsize zone [53]. We measured the total duration of time spent in the center zone (seconds), total vertical counts (i.e., rears, an additional index of exploration), the total ambulatory distance traveled (centimeters), and the average speed of ambulation (centimeters per second). Total vertical counts, known as rearing, is an exploratory behavior that suggests that animals are exhibiting increased locomotion and decreased anxiety-like behaviors [53]. 

### 2.12. Elevated Zero

After open field testing, behavior in the elevated zero maze was quantified, to further assess potential anxiety-like behavior; see review [54]. Mice were placed on an elevated circular platform that is approximately 50 cm tall and 60 cm in diameter. The platform contained alternating quadrants, with two quadrants enclosed by walls and two that were open. Mice were placed individually in an open quadrant, directly facing a closed quadrant, and were allowed to explore the circular platform for a total of five minutes. A camera mounted on the ceiling measured the total distance traveled and total time spent in the open quadrants using EthoVision XT software version 7.1.426 (Noldus Information Technology, Leesburg, VA, USA). Mice generally find the open quadrants less secure and more aversive, so less time spent in the open areas indicates more anxiety-like behavior, compared to spending more time in the open portions of the platform. 

### 2.13. Contextual Fear Conditioning

For the final behavioral test, we utilized a contextual fear conditioning paradigm to investigate the potential effects of different diet conditions on learning and memory in male mice, as described in our previous studies [55]. In this Pavlovian learning paradigm, mice were placed into an automated chamber containing an electrified grid floor, a polka-dot wall, and a peppermint olfactory cue (Coulbourn Instruments, Whitehall, PA, USA; 17.78 × 17.78 × 30.48 cm) for the training session. After an acclimation period of 120 s, mice received a 2-s, mild (0.5 mA) foot shock. Following the foot shock, mice remained in the chamber for an additional 60 s, before being transported back to their standard home cages. Twenty-four hours after the training session, mice were returned to the conditioning chamber, and freezing behavior, sans foot-shock, was measured for 120 s, utilizing FreezeFrame™ software version 3.19 (Actimetrics Software, Wilmette, IL, USA). Freezing behavior was assessed as the strength of the learned association between the footshock (UCS) and the conditioning chamber context (CS). In this behavioral paradigm, mice that learn well the association between the visual/tactile/olfactory context and the aversive footshock exhibit increased freezing behavior compared to mice that do not learn this association as well [56].

## 3. Results

### 3.1. Gut Microbiome Diversity Decreased with Consumption of the Synthetic Diet

Changes in diversity of the gut bacterial microbiome in the two feeding groups were assessed throughout the feeding period (Figure 2). Alpha diversity was determined through calculation of the Shannon diversity index (Figure 2A), and by counts of observed number of ASVs for each sampling point (weeks 0–13) (Figure 2B). At weaning (week 0), there was no significant difference in mean Shannon diversity (Synthetic x = 3.882, Standard x = 3.795, *t*-test *p* = 0.875). However, significant differences were detected after the first week compared to Week 0 and continued throughout the course of the study (Figure 2), such that by week 13, the synthetic diet group was significantly less diverse (Synthetic x = 2.485, Standard x = 4.595, *t*-test *p* ≤ 0.001). A similar result was obtained through counts of unique ASVs, with no difference found at weaning (Synthetic x = 205.333, Standard x = 202, *t*-test *p* = 0.959) and significantly less observed ASVs by week 13 (Synthetic x = 148.9, Standard x = 301.3, *t*-test *p* ≤ 0.001). 

The gut microbial community composition of the synthetic diet and standard diet animals at week 13 was compared using principal component analysis (PCA). Phylogenetic isometric log-ratio (PhILR) transformed distances was utilized to produce the PCA plots (Figure 3) [46]. At week 13, there is no overlap between samples in the two groups, indicating a dramatic shift in the composition of the gut microbiome caused by the synthetic diet. PERMANOVA determined that the impact of the deviance based on diet (synthetic vs. standard) was statistically significant (R^2^ = 0.56, *p* = 0.001), explained 56% of variance [48].

In order to identify phylogenetic groups of ASVs which were influential drivers in the variation between the two diet groups at week 13, phylofactor analysis was conducted (Figure 4) [49]. Phylofactorization involves all bacterial ASVs present in stool samples from mice fed either a synthetic or standard diet at week 13 (Figure 4A). Three factors (family clades) were identified as drivers of variation between the two diet groups: *Erysipelotrichaceae* was enriched in the synthetic diet group (purple; *p* < 0.001) (Figure 4B), *Prevotellaceae* was depleted in the synthetic diet group (blue; *p* < 0.001) (Figure 4C), and *Muribaculaceae* was depleted in the synthetic diet group (yellow; *p* < 0.001) (Figure 4D).

Following the characterization of shifts in microbial diversity and the gut microbiome composition associated with the synthetic diet, the next step involved exploring the metabolic pathways that might be affected by these variations. Metabolic predictions were carried out on microbiome data collected at week 13 of the feeding period using PICRUSt2 (Appendix A) [51]. Consistent with the substantial separation of gut bacterial profiles seen in the week 13 PCA (Appendix A), 46 pathways were predicted to be significantly altered in the synthetic diet group compared to the standard group (Welch’s inverted method with Bonferroni corrections for multiple comparisons: 46 pathways *p* ≤ 0.05) (Appendix A). Illustrated in Appendix A are the top 14 pathways (*p* ≤ 0.001). Shown in red are pathways enriched in the synthetic diet group and in blue are pathways enriched in the standard diet group. Possible functions affected include fermentation, sulfate reduction, and amino acid biosynthesis.

### 3.2. Shortened Colon Length in Synthetic Diet Group

Colon shortening was observed in the synthetic diet group despite there being no difference in body weight between different diet conditions (Figure 5A), mice fed the synthetic formula diet were found to have significantly shorter colons (cecum to anus). Synthetic diet animals (7.85 cm ± 0.65) had a dramatically shorter colon than that of standard diet animals (9.83 cm ± 0.48), *p* = 2.65 × 10^−7^ (Figure 5B). Photos of colons extracted from mice in the synthetic diet and standard diet group are presented in (Figure 5C).

### 3.3. Decreased Short Chain Fatty Acid Levels in Synthetic Diet Group

Short chain fatty acids are produced as primary terminal metabolites by multiple species of commensal gut bacteria in the colon. SCFA concentrations were measured in the colon tissue following the 13-week feeding period through gas chromatographic analysis normalized by sample mass (Figure 6). 

Acetic acid was significantly lower in the synthetic diet group than the standard diet group (7.43 ± 3.17 vs. 6.39 ± 8.58, one-tailed *t*-test *p* = 0.04). Propionic acid concentrations were also found to be significantly lower in the synthetic diet animals (2.31 ± 0.96 vs. 4.34 ± 2.02, one-tailed *t*-test *p* = 0.04). Butyric acid concentrations were significantly lower in the synthetic diet animal samples as well (1.09 ± 0.56 vs. 4.16 ± 2.04, one-tailed *t*-test *p* = 0.01).

### 3.4. Altered Cytokine Profiles in Synthetic Diet Group

Independent samples *t*-tests were performed to investigate the effects of consuming the synthetic diet on proinflammatory cytokines TNF-α and IL-6, and anti-inflammatory cytokine IL-10 in the serum, hippocampus, and colon (Figure 7). In the serum IL-6 analysis, homogeneity of variance was violated. Thus, two samples that were determined to be outliers by SPSS’s interquartile range rule were excluded from analysis. The results revealed a difference in serum IL-6 that approached significance (*t*(15) = 1.791, *p* = 0.093), such that the synthetic diet mice had serum levels of IL-6 that trended higher than that of control diet mice. The results revealed no significant differences between mice consuming the synthetic diet and control diet in terms of serum TNF-α (*t*(18) = −1.408, *p* = 0.176) or serum IL-10 (*t*(18) = −0.454, *p* = 0.655). 

Colon tissue analyses revealed no significant differences in levels of TNF-α (*t*(8) = −0.966, *p* = 0.362), IL-6 (*t*(8) = 0.466, *p* = 0.653), or IL-10 (*t*(6) = −1.149, *p* = 0.294) between synthetic diet mice and standard control diet mice. 

Conversely, the results of cytokine analyses in the hippocampus revealed that IL-6 levels (*t*(16) = 4.568, *p* < 0.001) were significantly higher in the hippocampal lysates of synthetic diet mice compared to control diet mice. TNF-α levels did not significantly differ between synthetic diet mice and standard diet mice (*t*(18) = 1.314, *p* = 0.205). Hippocampal IL-10 was unable to be assessed, as values from all mice consuming the synthetic diet and all but four mice consuming the standard diet fell below the standard curve of the assay.

### 3.5. Modestly Increased Ambulation in Synthetic Diet Group

Independent sample *t*-tests were performed to analyze the effects of the synthetic diet on anxiety-like behavior, exploratory behavior, and locomotor activity in the open field test (Figure 8). The results revealed no significant differences between the total time spent in the center zone (*t*(18) = 1.46, *p* = 0.16), total vertical counts, (*t*(18) = 0.96, *p* = 0.35), or total distance traveled (cm) (*t*(18) = 1.85, *p* = 0.08). Additionally, the results revealed a significant difference between the diet treatment groups in terms of the total average speed of ambulation (*t*(17) = 2.33, *p* = 0.03), such that mice on the synthetic diet traveled significantly faster than mice on the standard control diet. 

### 3.6. Synthetic Diet Had No Effect on Other Motor and Cognitive Functions

Independent sample *t*-tests were performed to analyze potential effects of the synthetic diet on motor and anxiety-like behaviors in the elevated zero maze (Figure 9). The results revealed no significant differences between time spent in open quadrants (*t*(18) = 1.52, *p* = 0.15), latency to first entry of an open quadrant (*t*(18) = 0.08, *p* = 0.94), number of entries to the open quadrants (*t*(18) = 0.90, *p* = 0.38), total distance traveled (*t*(18) = 1.01, *p* = 0.29), or velocity (*t*(18) = 0.96, *p* = 0.35) between mice that consumed the synthetic diet versus mice that consumed the standard control diet. 

Cognitive function was assessed using the contextual fear conditioning method. An independent samples *t*-test was also performed to assess the effects of the synthetic diet on learned freezing behaviors (Figure 10). The results revealed no significant difference in freezing behavior on testing day (*t*(18) = −0.26, *p* = 0.81) and training day (*t*(18) = −0.06, *p* = 0.96) between mice fed the synthetic diet and mice fed the standard control diet.

## 4. Discussion

The gut microbiome has been implicated in various aspects of health and disease as a result of the varied roles that resident bacteria play in nutritional absorption, metabolite production, education of immune cells, host protection, and interactions with distal organs (i.e., the gut-brain axis). Additionally, decreased gut microbiome diversity has been associated with a variety of disease states [57]. Moreover, diet is frequently reported as a major driving force behind microbiome composition. Here, we assessed the effects of a largely synthetic diet analogous to those typically prescribed for PKU patients and others with genetic inborn errors of amino acid metabolism. Wild-type mice were chosen for this study, rather than a metabolism disorder mouse model, to isolate the synthetic formula diet as a factor in the modulation of the gut microbiome previously seen in metabolic disorders such as PKU [36,58,59]. 

Western societies consume disproportionately high amounts of fiber-deficient processed food, whereas traditional societies consume more fiber. Consequently, traditional societies exhibit enhanced gut microbial diversity and have been shown to contain gut commensal bacteria no longer seen in industrialized societies [25,30,60]. Western diets also typically contain fewer microbially accessible carbohydrates (MACs), possibly leading to the loss of certain bacterial groups [30,31]. For example, the Western diet has been shown to decrease the population of beneficial bacteria while simultaneously increasing disease-associated bacteria in the gut [27,61]. Li and colleagues (2018) studied the effects of diet on inflammation and the gut microbiome in C57BL/6J mice and found that consumption of a Western diet induced a pro-inflammatory response, increased the relative abundance of potentially pathogenic microbes, such as *Escherichia coli*, and ultimately reduced microbial diversity. Conversely, this same study found that a plant-based diet fostered the production of healthy, beneficial gut bacteria, such as *Bifidobacterium* and *Lactobacillus* [27]. Additional studies in C57BL/6J mice have also demonstrated that high-fat, Western-style diets reduce the presence of *Bifidobacterium* and increase the population of bacteria associated with pro-inflammatory conditions [61]. Furthermore, gut microbial diversity plays a protective role by contributing to stability and resilience against major changes and is also associated with enhanced genetic and enzyme diversity [62,63,64]. In mice consuming an oversimplified and highly processed synthetic diet, we detected a drop in the Shannon diversity index and ASV counts after the first week of feeding. This pattern continued throughout the 13-week course of the study, such that these diversity measures for synthetic diet-fed mice were ultimately lower than their initial post-weaning baseline.

Comparative taxa abundance showed an increased presence of bacteria previously associated with disease and inflammation in the synthetic diet-fed mice. For example, *Prevotella* was significantly more abundant in the mice fed the synthetic diet compared to mice fed the standard diet. In a recent study, investigators reported that mice colonized with certain *Prevotella* species exhibited several markers of inflammation, including shortening of the colon, increased pro-inflammatory IL-6 in the gut, and decreased anti-inflammatory IL-10, similar to the findings from the present study for mice in the synthetic diet group [65]. 

A representative PCA plot of the synthetic diet and standard diet microbiome data illustrates the dramatic shift in the microbiome over the course of the feeding period (Figure 3). This shift can be visualized based on the distances between data points representative of each feeding group and verified through PERMANOVA (48). Our results indicated that the difference between the microbiome data of the two diet groups was statistically significant, and that 56% of the variance could be explained by the diet group assignments. 

Metabolic predictions were made by combining the 16S rRNA phylogenetic marker gene data with open sources of known bacterial pathways from whole-genome sequences of closely related strains (Appendix A). Our results demonstrated that abundances of bacteria associated with several pathways were affected by the consumption of a synthetic diet. Many of the pathways highlighted through the predictions contribute to fermentation, as well as sulfate reduction and amino acid biosynthesis. The colon is the primary site for the anaerobic fermentation of indigestible dietary fibers that provide SCFAs, including acetate, propionate, and butyrate [28]. SCFAs have been shown to be protective against colitis and general inflammation in the gut, decreasing the colonic pH, neutralizing toxins, and preventing the adhesion of pathogenic microbes in the gut [66,67,68,69,70].

A dramatic decrease in colon length was found at the time of tissue collection in this study. Further studies should be conducted to elucidate whether this may be due to inflammation in the colon or due other factors such as dietary fiber. This shortening of the colon has possible implications for nutritional absorption and could be, in part, caused by the inability of nutritionally depleted colonocytes to enter a maintenance phase. In this study, significantly lower SCFA levels, specifically acetic acid, propionic acid, and butyric acid, were also found in the colonic contents of mice consuming the synthetic diet. Butyrate in particular is known to be a major energy source for colonocytes and is produced as a result of the fermentation of soluble fibers by resident colonic bacteria [28,71,72]. Therefore, the consumption of dietary fibers is believed to result in the increased production of SCFAs, allowing colonocytes to enter the maintenance phase, supporting the structure of tight junctions, and enhancing the gut barrier [2,73,74,75]. Multiple studies have shown improvement in symptoms in several diseases and disorders with the addition of SCFAs or dietary fiber [10,20,25,32,72,76,77,78].

Mice that consumed the synthetic diet exhibited only minimal alterations in behavior. Mice, in order to evade predation, will typically avoid open areas, staying close to walls or cover. These types of fear behaviors are typically considered anxiolytic [79]. Interestingly, mice that consumed the synthetic diet in this study exhibited significantly higher average speeds in the open field test and traveled moderately farther than those on the standard diet, suggesting that the synthetic diet modestly increased locomotor activity, generally. However, diet did not affect emotionality or learning/memory in the behavioral tasks employed in the current study. There were no indicators of elevated anxiety levels in the open field or the elevated zero maze tasks for either treatment group, and no difference in freezing behavior (indicative of contextual memory) between mice consuming the synthetic diet and their conspecifics consuming the standard control diet during contextual fear conditioning. 

In conclusion, we demonstrated that the consumption of a synthetic diet led to the rapid emergence of indicators of dysbiosis in the gut microbiota, as well as possible markers of inflammation in multiple tissues. Interestingly, though the synthetic diet tested here cannot be considered high-fat as compared to a typical Western diet, mice on the synthetic diet showed results similar to those associated with the typical Western diet, including increases in multiple potential markers of inflammation and concomitant losses of anti-inflammatory bacteria [19,25]. Given that the synthetic diet used in the current study contains less soluble and insoluble fiber than the standard diet of the controls, the results suggest that the reported effects of the Western diet may be more strongly influenced by the lack of microbially accessible fiber than increased fat or animal protein intake. The findings reported here also have implications for the consumption of restrictive diets, such as those utilized in medical interventions for patients with PKU or other genetic inborn errors of metabolism. Although more research is necessary to untangle the individual effects of the replacement of complex proteins with amino acids, versus decreases in complex carbohydrates, these data suggest that such diets should also consider microbially accessible carbohydrate fibers as important components.

## Figures and Tables

**Figure 1 microorganisms-11-02694-f001:**
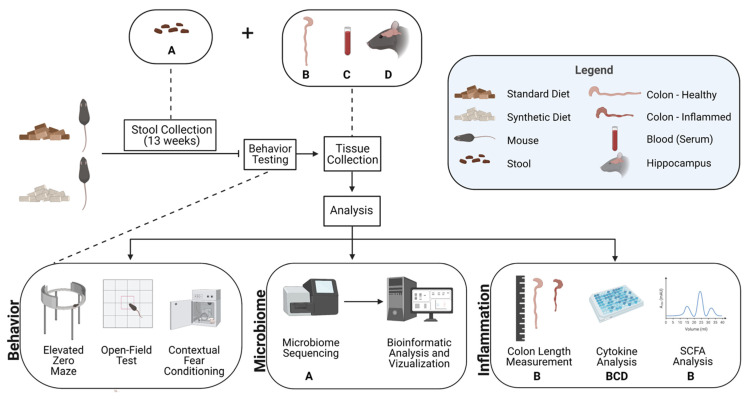
Diagram of experimental design to investigate the effects of consuming a synthetic formula diet on behavior, gut microbiome, and inflammation. Tissue types are denoted by illustrations described in the legend and represented by a letter beneath the assay in which they were examined (A—Stool; B—Colon; C—Blood (serum); D—Hippocampus).

**Figure 2 microorganisms-11-02694-f002:**
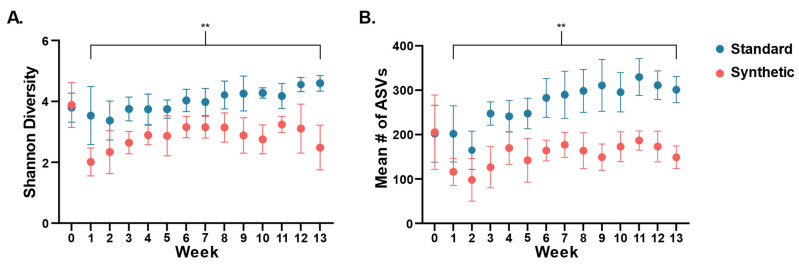
Divergence of gut microbiota alpha diversity ((**A**): Shannon diversity index; (**B**): number of observed ASVs) based on assigned diet plotted from week 0 (weaning) through week 13. Dots indicated mean values (*n* = 10 per group). Error bars were standard deviation of the mean. (** *p* < 0.01).

**Figure 3 microorganisms-11-02694-f003:**
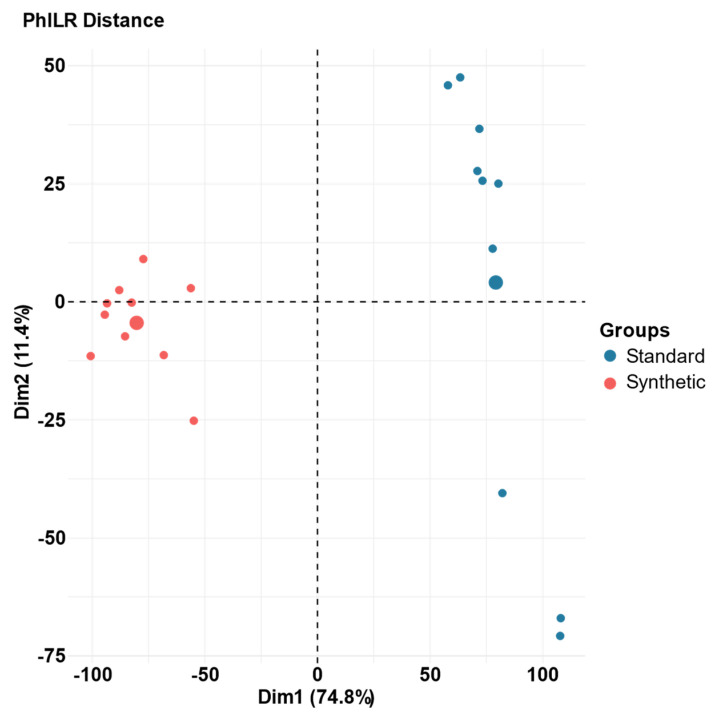
Principal component analysis (PCA) of PhILR distances for the gut microbiota of synthetic and standard diet animals after 13 weeks of feeding on the assigned diet. Individual samples are represented by one point; larger circles represent the mean coordinates for each diet group (*n* = 10 per group). (R^2^ = 0.56, *p* = 0.001).

**Figure 4 microorganisms-11-02694-f004:**
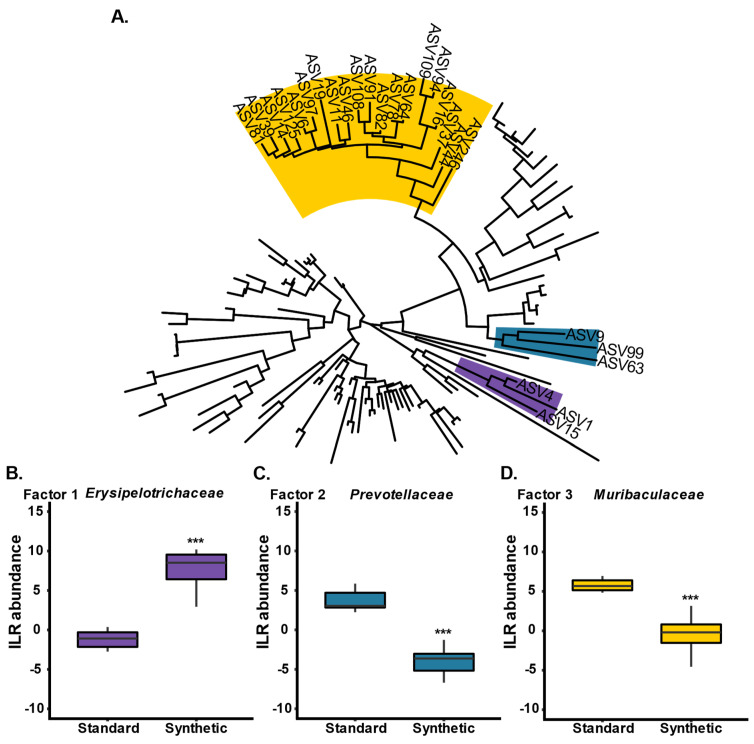
Phylofactorization of all bacterial ASVs present in stool samples from mice fed either a synthetic or standard diet at week 13 (*n* = 10 per group) (**A**). Three factors (family clades) were identified as drivers of variation between the two diet groups: (**B**) *Erysipelotrichaceae* (purple), (**C**) *Prevotellaceae* (blue), and (**D**) *Muribaculaceae* (yellow). (*** *p* ≤ 0.001).

**Figure 5 microorganisms-11-02694-f005:**
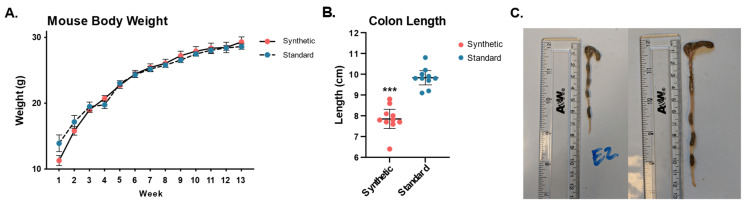
Mouse body weights and colon length measurements of mice fed the synthetic diet or the standard diet after 13 weeks (*n* = 10 per group). Body weight for all mice was measured weekly throughout the 13-week feeding period (**A**). Synthetic diet animals (7.85 cm ± 0.65) had a significantly shorter colon than that of standard diet animals (9.83 cm ± 0.48), *p* = 2.65 × 10^−7^ (**B**). Representative photos of colons dissected from a synthetic diet animal (**C**—left) and a standard diet animal (**C**—right). (*** *p* ≤ 0.001).

**Figure 6 microorganisms-11-02694-f006:**
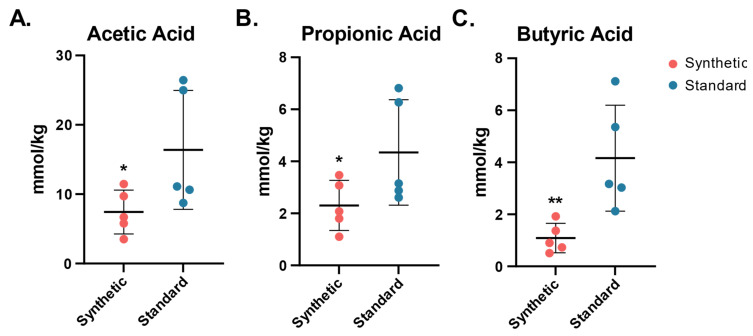
SCFA concentrations (mmol/kg) in colon tissue of mice fed the synthetic diet or the standard diet after 13 weeks (*n* = 5 per group). Dots represent the concentration of SCFA per mouse, the center bar indicates the standard mean concentration within group (synthetic diet, standard diet). (**A**) Acetic acid synthetic mean ± SDSEM (7.43 ± 3.17), standard mean + (16.39 ± 8.58), *t*-test one tail *p* = 0.04; (**B**) Propionic acid synthetic mean + (2.31 ± 0.96), standard mean + (4.34 ± 2.02), *t*-test one tail *p* = 0.04; (**C**) Butyric acid synthetic mean + (1.09 ± 0.56), standard mean + (4.16 ± 2.04), *t*-test one tail *p* = 0.01. (* *p* ≤ 0.05; ** *p* ≤ 0.01).

**Figure 7 microorganisms-11-02694-f007:**
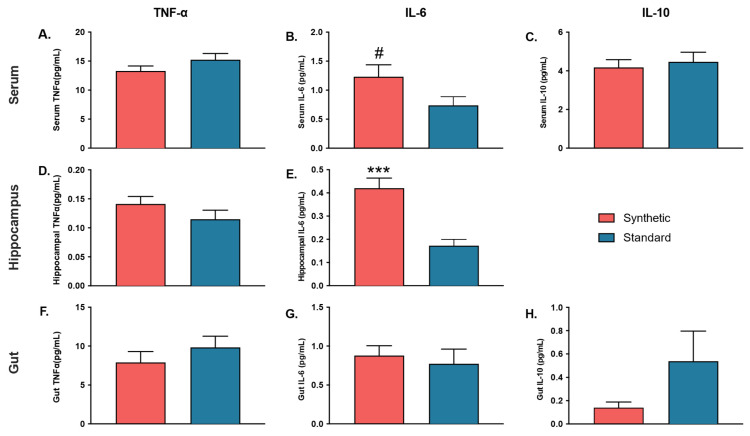
Cytokine (TNF-α, IL-6, IL-10) quantification in the serum (*n* = 10 per group), hippocampus tissue lysate (brain) (*n* = 10 per group), and colon tissue lysate (gut) (*n* = 5 per group). (**A**) Serum TNF-α; (**B**) Serum IL-6; (**C**) Serum IL-10; (**D**) Hippocampal TNF-α; (**E**) Hippocampal IL-6; (**F**) Gut TNF-α; (**G**) Gut IL-6; (**H**) Gut IL-10. Hippocampal IL-10 was excluded. Bars represent ± SEM. # *p* ≤ 0.10; (*** *p* ≤ 0.001).

**Figure 8 microorganisms-11-02694-f008:**
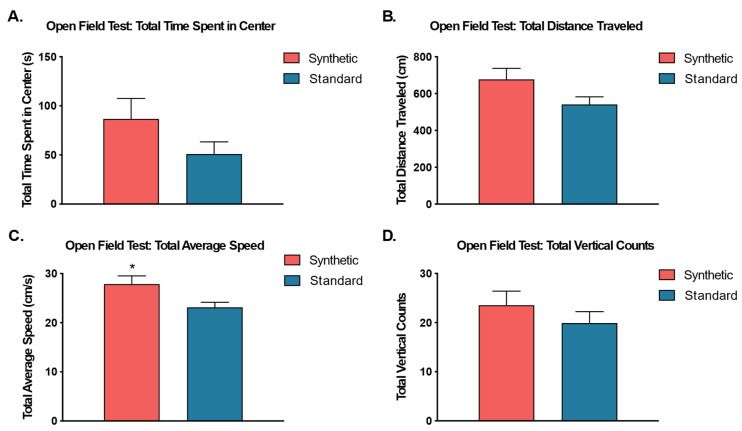
Open Field Test. An independent samples *t*-test was used to compare behavior in the open field chamber between animals fed synthetic diet (coral) and a standard diet (blue) (*n* = 10 per group) for (**A**) total time spent in the center zone, (**B**) total distance traveled, (**C**) total average speed, and (**D**) total vertical counts. Although all cases revealed higher average values for synthetic diet fed mice, only total average speed (**C**) was significantly different. (* *p* ≤ 0.05).

**Figure 9 microorganisms-11-02694-f009:**
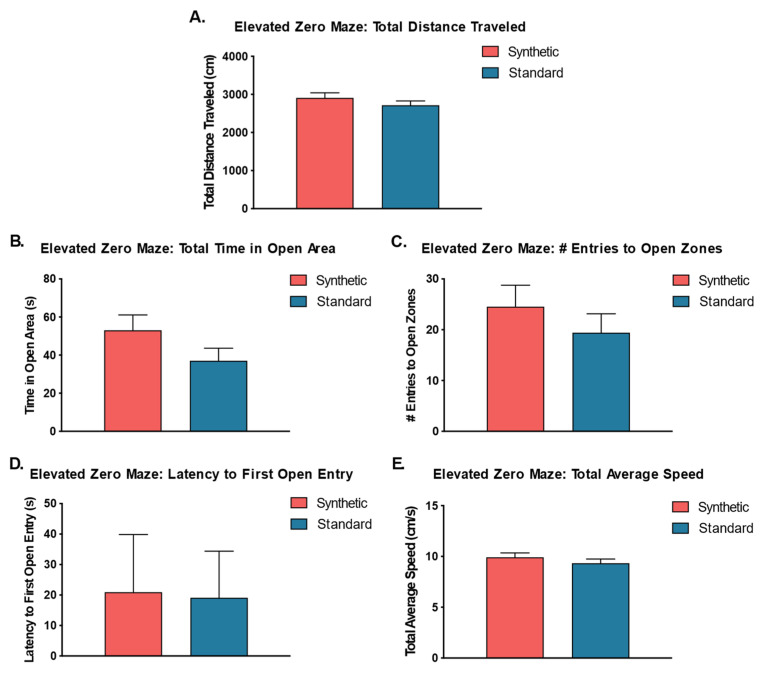
Elevated Zero Maze (*n* = 10 per group). (**A**) Total distance traveled in synthetic and standard diet animals; (**B**) Total time in open areas; (**C**) Number of entries into the open areas or zones; (**D**) Latency to first entry into an open zone; (**E**) Total average speed. No significant differences were detected by independent samples *t*-tests.

**Figure 10 microorganisms-11-02694-f010:**
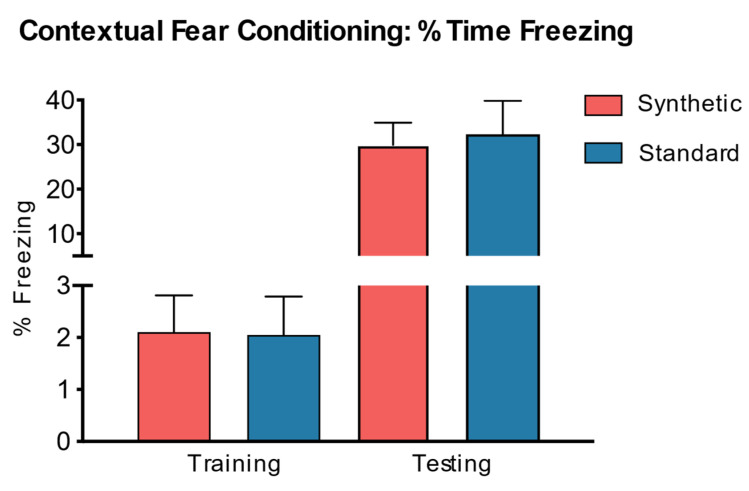
Contextual Fear Conditioning. Independent samples *t*-test revealed no significant difference in percent freezing time (indicates contextual memory on the training day) between mice on a synthetic diet and standard diet (*n* = 10 per group) following behavioral testing (Training: *t*(18) = −0.06, *p* = 0.96, *n* = 10; Testing: *t*(18) = −0.26, *p* = 0.80, *n* = 10).

## Data Availability

The datasets for this study can be found in the National Library of Medicine [https://www.ncbi.nlm.nih.gov/bioproject/908095].

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
