# Peer review of "A Synthetic Formula Amino Acid Diet Leads to Microbiome Dysbiosis, Reduced Colon Length, Inflammation, and Altered Locomotor Activity in C57BL/6J Mice"

_microorganisms, 2023, doi:10.3390/microorganisms11112694_

Round 1

Reviewer 1 Report

Comments and Suggestions for Authors

Authors compared two different diets in mouse models, resulting in lower microbial diversity, reduced colon length, inflammation, and altered locomotor activity in the experimental group compared to the control group. While the findings are intriguing, there are areas that require improvement. Here are some examples: 

1: The abstract is not well written it is not an informative abstract covering the problem, the aim, methodology, a proper conclusion, The initial paragraphs (first 11 lines (1 to 11)) of abstract is more like Introduction, an abstract should provide a concise summary of the study's key points without exploring into background information. 

2: There is finding relevant to the 16S rRNA finding in the abstract (except lower diversity mentioned), authors may include a statement on the main variation (increased or decreased) microorganisms after this diet in comparison with control group.  

3_ The phrase "may render other health consequences" is unclear. Consider rephrasing it for better clarity, such as "may have additional health… 

4_ The abstract should briefly mention important methodological aspects. Please consider adding a sentence that provides information about the number of mice used and the primary methodology. For example, you could say something like, "Using a cohort of [number] male/female C57BL/6J mice and a [briefly describe methodology], we investigated..." 

5_ The authors need to check the referencing style of the journal, this style is not recommended by the journal. Even the current style is not consistent; Some include author names and publication years, while others don't. Use a consistent citation style throughout the text. Please use what is recommend by the journal should be numbered style.  

6_While the introduction provides a comprehensive overview of the importance of the gut microbiome and the role of dietary fiber, it can be shortened for brevity. Consider summarizing key points to maintain the reader's interest. 

7-The introduction provides valuable background information but doesn't effectively introduce the focus of the study. Consider adding a sentence that clearly states the research question, study aims, or the purpose of the study to fill the current research gap. 

8- You have stated that care followed the Guide for the Care and Use of Laboratory Animals, but consider mentioning the specific guidelines or protocol numbers followed for ethical approval. 

9- Authors may consider ending the introduction with a sentence that summarizes the significance of the study and emphasizes the research gap or unanswered questions in the field. 

10-Authors mentioned that male C57BL/6J mice were used, It's important to add more specific details, such as the age, weight, and any other relevant characteristics of the mice used.  

11- Regarding housing condition authors may want to include more information about the room temperature, humidity, bedding material, and other factors that could affect the study. 

12-Please explain the criteria for assigning mice to each group (e.g., randomization, matching criteria). 

13- Mouse fecal samples were collected weekly for gut microbiome analysis. It is worth pointing out briefly the rationale behind this sampling frequency. 

14- Briefly explain methods or procedures used for tissue collection and preservation  

15- Please specify parameters or criteria used for trimming and quality filtering, techniques for tissue extraction,

Reviewer 2 Report

Comments and Suggestions for Authors

This article briefly introduces the rich fiber diet is beneficial to the ecological development of the microbial group in mammals maintain physiological balance, and discusses the synthetic formula diet because of excessive refinement and lack of fiber intake, lead to experimental mice microbiota biodiversity reduction and balance imbalance, and by comparing the synthetic formula diet and standard diet behavior patterns of mice, metabolites and inflammatory markers level indicators, get the synthetic formula diet of mice exercise, cognitive, immune and other functions. It is interesting work. However, there still have some issues need to check:

1.        The general article is not divided into chapters, and the structure is more disordered.

2.        Line47-49. The background about gut microbiome should be updated in recent years. Please refer this reference (Food Chemistry, 402(2023): 134231; Critical Reviews in Food Science and Nutrition, 63(19): 3895-3911).

3.        In this paper, only four experimental mice are used for control, the number is too small, so the experiment is not convincing. It is recommended that at least six mice are in one group, with two groups for each control. Please refer this reference (Food Bioscience, 50(2022): 101946; Food & Function, 2022, 13(24), 12686-12696).

4.        The supplementary table mentioned in page 4 does not appear in the article and needs to be supplemented.

5.        Page 17, lines 300 and 309“TNFα”should be “TNF-α”many times, in the text, the literature citation format is not unified. Please refer other references about cytokine (Cancer Research, 2020, 80(12): 2564-2574).

6.        It is recommended to use digital annotation, so that it is easy to find the source of content reference.

7.        In the article, only half of the error bars in the last four pictures are shown, the last picture is hasty, and the English name format of the coordinate axis is confused.

8.        The reference format is wrong, and there are too many capitals in the English names.

Comments on the Quality of English Language

This article briefly introduces the rich fiber diet is beneficial to the ecological development of the microbial group in mammals maintain physiological balance, and discusses the synthetic formula diet because of excessive refinement and lack of fiber intake, lead to experimental mice microbiota biodiversity reduction and balance imbalance, and by comparing the synthetic formula diet and standard diet behavior patterns of mice, metabolites and inflammatory markers level indicators, get the synthetic formula diet of mice exercise, cognitive, immune and other functions. It is interesting work. However, there still have some issues need to check:

1.        The general article is not divided into chapters, and the structure is more disordered.

2.        Line47-49. The background about gut microbiome should be updated in recent years. Please refer this reference (Food Chemistry, 402(2023): 134231; Critical Reviews in Food Science and Nutrition, 63(19): 3895-3911).

3.        In this paper, only four experimental mice are used for control, the number is too small, so the experiment is not convincing. It is recommended that at least six mice are in one group, with two groups for each control. Please refer this reference (Food Bioscience, 50(2022): 101946; Food & Function, 2022, 13(24), 12686-12696).

4.        The supplementary table mentioned in page 4 does not appear in the article and needs to be supplemented.

5.        Page 17, lines 300 and 309“TNFα”should be “TNF-α”many times, in the text, the literature citation format is not unified. Please refer other references about cytokine (Cancer Research, 2020, 80(12): 2564-2574).

6.        It is recommended to use digital annotation, so that it is easy to find the source of content reference.

7.        In the article, only half of the error bars in the last four pictures are shown, the last picture is hasty, and the English name format of the coordinate axis is confused.

8.        The reference format is wrong, and there are too many capitals in the English names.

Reviewer 3 Report

Comments and Suggestions for Authors

General comments:  The effects of free amino acids vs protein in the diet is an interesting question. You have collected a wide range of data on the mice that will be of some interest to readers. I was especially pleased to see the combination of behavioural phenotypes, microbiome charcaterization and intestinal physiology. Unfortunately, the study design here is intrinsically limiting with respect to interpretation which means the significance of new insights is constrained. In my opinion there is a tendency to overstate the findings.

Avoid hyperbole thoughout.  In numerous cases you say 'essential' - which grammatically means "can not occur without" when you really mean important or beneficial.  The third line of the abstract is an example - dietary fibre is beneficial in resisting infection but not essential.

Many interpretations are written in ways that imply evidence for causality that simply isn't there. You have a stratified study design (two treatment groups), with small number of animals (and unclear how many are independent replicates - separately caged), and the variable you aim to test, amino acid vs protein, is not the only factor manipulated (you actually manipulate every component of the diet!  This occurs throughout and first example is in abstract - Line 34 states "this synthetic diet altered locomotor activity".  You observed a (small) difference in locomotor activity in the ten mice on synthetic diet relative to the ten mice on standard diet. The diets differed in many uncontrolled factors and the experiment was not repeated. The observation is worth reporting but please don't state it as though it implies mechanistic causality.

Specific comments.

L55-58. The description of microbiome influence on cognition and behaviour overstates that evidence for this. 

L60-62. This is rather vague - at least the way the Muegge et al paper defined the meaning of 'functional "core" community' should be described.

L90. Do these studies actually show this?  The claim that reduced fibre induces inflammation implies a controlled study where participants were followed longitudinally and inflammation was induced in response to fibre intake specifically being reduced.  Phrase statement more carefully if they don't.

L98. 'provide gut microbiota with essential anti-inflammatory nutrients' is misleading.  The microbes are not using the molecules that have anti-inflammatory activity as nutrients.

L99. The Firmicutes-Bacteroidetes ratio has repeatedly been shown to be an uninformative generalization.  Most recently see Walker and Hoyles (2023) Nature Microbiology

L108. The concept of dysbiosis is widely misused in the literature - here it would be better to say increases risk of dysbiotic host interaction developing.

L118. This does not seem a relevant reference for the statement.

L140. 'similar bacterial functional groups' is not informative - if you mean butyrogenic bacteria - and there is actually evidence for this - say so.

L172. Please also state the number of cages earlier you say 4 animals per cage - which does not obviously match to 10 animals.

L175-182. The major treatment for th estudy is diet - yet it is very difficult for a reader to actually see what was involved.  Either put a Table in or describe more effectively. It appears that there was no single diet component that was in common between the two diets - carbs fat and protein all differed.  It is unclear if the amino acid distribution was matched.  It is not explained that the synthetic diet has cellulose as the sole source of 'fibre', or that this is essentially unavailable to microbes in mice.

L203-271. The first three methods sections are fine, but have a lot more detail than necessary.  They could be reduced to save space if desired.  I query if the analysis pipeline is fully described.  It would be unusual to have only ca 200 ASV's from the pipeline as described.  Typically there would be more than a thousand 'rare' ASvs the bulk of which are removed by abundance-ubiquity filtering steps to leave about 200.  I presume such a filtering step was done here but it is not described.

L273. Was the colon empty of gut contents?  Process for extruding stool and flushing the colon should be described if performed. Presence of other material besides colon tissue acknowledged if not.

L285 (and results) Measuring SCFA in the colon tissue itself is unusual choice.  The rationale for this should be explained and taken into account when interpreting.

L396-419. The DESeq and Phylofactor are basically addressing the same question.  It is not necessary to show both figure sin main text and this could be described more concisely. Presentation of DESeq2 analysis in Fig 4 is not helpful to reader.  What does it mean to show the same taxon as both under- and over-represented?

L420. Lets be honest - PICRUSt2 is a pretty weak tool.  It should not be presented as though it is a 'real' dataset because it is essentially autocorrelated to the taxon-based analysis.  Its only value is to test hypotheses as to what genomic traits may be over-represented across the set of enriched taxa.  This can be described more simply and the Figure 6 data is both uninformative as presented and best relegated to supplementary.  

L440. Colon length may be reduced in cases of chronic inflammation such as colitis models.  However, this is NOT the same as saying colon length is a proxy for inflammation.  The gut of most mammals i s aplastic tissue and length/volume changes with many parameters other than inflammatory state (notably diet fibre, fecal 'bulking').

L456-7. The physiological meaning of fermentation precludes any possibility other than 'anaerobic' and the SCFA are not 'byproducts' but the primary terminal metabolites.  In the mouse the vast majority of this occurs in the cecum not the colon.  The colon tissue is not the most physiologically relevant site to explore, although SCFA are expected to have effects there.

L499+. Th eextensive description of the behavioural tests is not the most exciting read.  The inclusion of these is a strength of the study, but they could be presented far more concisely in my opinion given that almost all show no effect.  Also not clear if all 10 mice were assessed or how many indpendent observation on each mouse were made.

L562. I don't think you can make a case for any gut bacteria being 'essential' and 'harmful Gram-negative' is uninformative.

L565-7. The phrases 'production of harmful/healthy microbes' are grammatically incorrect - you likely mean increase in relative abundance.

L594+. This not presented accurately.  You measured change in the bacterial taxa then inferred genome composition of those taxa.  The pathways are not affected - the cell abundances are.

L623 and 627. It is an overstatement to say mice on synthetic diet exhibited alterations in behaviour, when only one measure showed a difference and the level of independent replication is unclear.

Comments on the Quality of English Language

English expression is generally good, but use of scientific concepts is imprecise.

Round 2

Reviewer 1 Report

Comments and Suggestions for Authors

NA

Reviewer 2 Report

Comments and Suggestions for Authors

It can be accepted in the current revision. 

Comments on the Quality of English Language

It can be accepted in the current revision. 

Reviewer 3 Report

Comments and Suggestions for Authors

Generally the manuscript has improved and responses to my comments are all fair.

I think you have misunderstood regarding the DESeq issue.  In my opinion it is of no help to a reader to show multiple data points for each ASV within a higher taxon.  You have moved the figure to supplementary data and at least now explain that the individual data points in the same row are distinct ASVs.  However the figure is still uninterpretable - we know some ASVs give a negative change and some a positive, but there is no way of knowing which is which.  
